# Robust Estimation of Deformation from Observation Differences Using Some Evolutionary Optimisation Algorithms

**DOI:** 10.3390/s22010159

**Published:** 2021-12-27

**Authors:** Mehmed Batilović, Radovan Đurović, Zoran Sušić, Željko Kanović, Zoran Cekić

**Affiliations:** 1Faculty of Technical Sciences, University of Novi Sad, Trg Dositeja Obradovića 6, 21101 Novi Sad, Serbia; mehmed@uns.ac.rs (M.B.); kanovic@uns.ac.rs (Ž.K.); 2Faculty of Civil Engineering, University of Montenegro, Bulevar Džordža Vašingtona bb, 81000 Podgorica, Montenegro; radovandj@ucg.ac.me; 3Institute of Architecture and Construction, South Ural State University, Lenin Prospect 76, 454080 Chelyabinsk, Russia; zcekic@singidunum.ac.rs; 4Faculty of Informatics and Computing, Singidunum University, Danijelova 32, 11000 Belgrade, Serbia

**Keywords:** robust M estimation, robust deformation analysis, evolutionary optimisation algorithms, Monte Carlo simulations

## Abstract

In this paper, an original modification of the generalised robust estimation of deformation from observation differences (GREDOD) method is presented with the application of two evolutionary optimisation algorithms, the genetic algorithm (GA) and generalised particle swarm optimisation (GPSO), in the procedure of robust estimation of the displacement vector. The iterative reweighted least-squares (IRLS) method is traditionally used to perform robust estimation of the displacement vector, i.e., to determine the optimal datum solution of the displacement vector. In order to overcome the main flaw of the IRLS method, namely, the inability to determine the global optimal datum solution of the displacement vector if displaced points appear in the set of datum network points, the application of the GA and GPSO algorithms, which are powerful global optimisation techniques, is proposed for the robust estimation of the displacement vector. A thorough and comprehensive experimental analysis of the proposed modification of the GREDOD method was conducted based on Monte Carlo simulations with the application of the mean success rate (MSR). A comparative analysis of the traditional approach using IRLS, the proposed modification based on the GA and GPSO algorithms and one recent modification of the iterative weighted similarity transformation (IWST) method based on evolutionary optimisation techniques is also presented. The obtained results confirmed the quality and practical usefulness of the presented modification of the GREDOD method, since it increased the overall efficiency by about 18% and can provide more reliable results for projects dealing with the deformation analysis of engineering facilities and parts of the Earth’s crust surface.

## 1. Introduction

The basic strategy of geodetic network optimisation entails minimising selected objective functions that are independent of the datum of the geodetic network. In this way, the problem is solved iteratively in a convergent process, where the current solution is better than the previous one. In the process of designing geodetic networks, a criterion matrix is used, which represents the required network quality so that the optimisation problem is solved directly. On the other hand, one of the most important tasks in the deformation analysis of geodetic networks is the selection of the optimal datum solution for the parameters of geodetic networks. Among deformation analysis methods that have been researched for theoretical and practical applications, it is important to mention the methods of conventional deformation analysis (CDA) [1,2,3,4,5,6,7,8,9,10,11,12,13] and methods of robust deformation analysis (RDA) based on M estimation [14,15,16,17,18,19,20,21,22], M_split_ estimation [23,24,25,26,27,28,29,30,31,32] and R estimation [33,34,35]. In conventional deformation analysis, it is essential to identify stable datum points, i.e., potential reference points (PRPs) of the network stabilised outside the zone of expected deformations. On the other hand, RDA methods are robust to the existence of displaced points in the potential reference part of the network, which makes them more convenient and simpler to apply than CDA methods. A number of studies can be found in the literature that analyse the efficiency of RDA methods in identifying displaced points by applying Monte Carlo simulations, and they have shown that these methods can be a significant alternative to CDA methods.

Deformation measurements of engineering facilities such as dams, bridges, tunnels, towers, etc., are mostly taken over short periods, except in cases when it is extremely important to monitor the object in a continuous manner. Reference [36] provides an overview of geodetic and global navigation satellite system (GNSS) sensors in order to carry out precise dam measurements. This review paper also presents the possibilities of terrestrial laser scanning, ground-based InSAR and advanced spaceborne DInSAR technology for monitoring the displacement and deformation of engineering facilities, in addition to traditional geodetic techniques and conventional deformation analysis. In the case of short-period measurements, it is possible to use identical equipment and observation plans in all measurement epochs. Hence, it is interesting to analyse the effects of constant errors that classical RDA methods based on M estimations, such as iterative weighted similarity transformation (IWST) [14] and its modifications based on the introduction of different optimisation conditions of robust estimation [15,19], cannot completely eliminate. Therefore, in the case of short-period measurements, the application of an alternative method called generalised robust estimation of deformation from observation differences (GREDOD) [18,19] has been proposed, because it completely eliminates the influence of systematic errors that burden the measurement results in certain epochs. In RDA methods based on M estimations, such as IWST and GREDOD methods, there are some shortcomings related to the statistical significance test of displacement. Consequently, there have been studies that aimed to overcome these problems [21].

RDA methods are widely used in the deformation analysis of geodetic networks. For example, the IWST method was used for the deformation analysis of the geodetic network of the Tevatron atomic particle accelerator complex at the Fermilab laboratory in the USA [37], which consists of about 2000 points. This method has also been implemented in the automated monitoring system “ALERT” developed by the Canadian Centre for Geodetic Engineering [38] and in GeoLab software for geodetic computation [39]. Furthermore, Reference [40] presents the application of the IWST method in combination with fibre optic sensors.

The selection of the optimal datum solution, regardless of the applied method, is a key step in the deformation analysis of geodetic networks, always with the aim of obtaining objective results. This primarily refers to geodetic networks of larger areas, where there may be diverse and dispersive causes of displacement in the geological sense, which may make it difficult to obtain an objective estimation. In this context, it is important to mention metaheuristic optimisation algorithms, such as simulated annealing (SA) and Hooke–Jeeves (HJ) algorithms, which can be used to identify a group of stable reference points, i.e., the optimal datum solution. Reference [41] showed that SA and HJ algorithms significantly contributed to the final decision in the process of identifying a group of stable reference points. In addition, the application of evolutionary optimisation algorithms—more precisely, the genetic algorithm (GA) and generalised particle swarm optimisation (GPSO) algorithm—in the robust estimation of the displacement vector in the IWST method was proposed and explained in Reference [22]. This significantly improves the efficiency of identifying displaced points on an object.

This paper analyses the GREDOD method, which represents a more recent alternative to the IWST method, from the aspect of applying some evolutionary optimisation algorithms in the robust estimation of the displacement vector. The deformation analysis procedure in the GREDOD method takes place in two phases: the robust estimation of the displacement vector and stability analysis of network points. The procedure of the robust estimation of the displacement vector comes down to solving the optimisation problem that is formed by the appropriate deformation model and objective functions. By solving the optimisation problem of the GREDOD method, the optimal weight values of the network PRPs, i.e., the datum points of the network, are determined. The values of these weights represent the contribution of each PRP to the datum definition of the displacement vector. 

This optimisation problem can be seen as a problem of determining the optimal datum of the displacement vector. In order to solve it, the iterative reweighted least-squares (IRLS) method [42] from the group of deterministic optimisation procedures is usually applied. The IRLS method starts from one initial solution obtained by the least-squares method and iteratively improves it during the optimisation process. However, if the initial solution is not in the vicinity of the real solution, it is not possible to determine the global optimal solution of the optimisation problem of the GREDOD method by applying the IRLS method [43]. 

Since the least-squares method is very sensitive to deviations from the model assumptions [44], it is quite clear that the initial solution can be significantly far from the real solution if the set of PRPs of the network contains displaced points, because in that case, the displacement vector follows a contaminated normal distribution. The distance of the initial from the real solution increases with the increasing contamination of the normal distribution, i.e., with an increase in the number of displaced network PRPs. Therefore, the efficiency of the IRLS method decreases with the increasing number of displaced PRPs, while the error of determining the optimal datum of the displacement vector increases.

In this context, it is evident that the optimisation problem of the GREDOD method should be considered from the global optimisation point of view. Consequently, this paper proposes the application of the GA and GPSO algorithms in the process of robust estimation of the displacement vector in the GREDOD method. It is important to point out that these algorithms search the solution space in a controlled random manner, which allows them to "jump out" of the local optimum in order to find the global optimum. The efficiency of applying the GA and GPSO algorithms in the robust estimation of the displacement vector in the GREDOD method is analysed using the mean success rate (MSR) based on Monte Carlo simulations. The results presented in this paper show that a significant increase in the robustness of the GREDOD method can be achieved using the proposed optimisation algorithms. This can certainly provide more reliable results when using the GREDOD method in practical applications in the area of geodetic monitoring of engineering facilities.

## 2. Materials and Methods

### 2.1. Genetic Algorithm and Generalised Particle Swarm Optimisation

Among a variety of global optimisation methods, a group named evolutionary algorithms is very popular and has often been applied in recent engineering problems. These methods are inspired by processes in nature and are known to be very robust and capable of finding the global optimum among a number of local optima. The genetic algorithm and particle swarm optimisation (PSO) are typical representatives of this group and are widely referred to in the literature [45,46].

The genetic algorithm is based on Charles Darwin’s theory of the evolution of species. The original form of the algorithm was created by John Holland in the 1970s [47], and its widespread application in real-world problems started with the expansion of computer techniques. The main instance in the algorithm is the so-called individual, which represents a potential solution, i.e., a set of variables defined by an optimisation problem. Each variable in the individual is referred to as a gene. The optimisation criterion represents a “fitness function”, meaning that individuals (variable sets) with a better value of the optimisation criterion are considered more fit. A predefined number of individuals creates the population. The initial population is randomly generated, and it is modified from iteration to iteration, called generations, using evolution mechanisms of selection (for selecting the best individuals for reproduction), crossover (combining genes, i.e., the variable values of selected individuals) and mutation (slight changes in the genes of individuals selected with a very small probability) in order to obtain the fittest individual, which represents the solution to the optimisation problem. Evolution mechanisms are determined by the set of parameters, which is described in more detail in the literature [48,49]. The entire procedure is described by the pseudocode of the genetic algorithm presented in Algorithm 1.

**Algorithm 1:** Pseudocode of genetic algorithm.**begin**
  k←0
  *generate initial population (randomly created set of individuals)*
  *calculate fitness (objective function value) for each individual*
  **while not** stopping criterion **do**
    k←k+1
    *select two individuals from the old generation for crossover*
    **bool** test = *probability test for crossover*
    **if** test **then**
      *perform crossover of two individuals to create two new individuals*
    **else**
      *new individuals = old individuals*
    **end**
    **bool** test = *probability test for mutation*
    **if** test **then**
      *perform mutation of new individuals*
    **end**
    *insert new individuals into population replacing old ones*
    *calculate fitness (objective function value) for new individuals*
  **end**
**end**

PSO is inspired by patterns of the social behaviour of animals living in groups, such as birds, fish or insects [50]. Contrary to the GA, which mimics the evolution of individuals, PSO is interpreted as the motion of individuals in the search space over time. Similar to the GA, each individual (called a particle) is determined by a set of variables’ values, which is interpreted as the “position” y of the particle, i.e., its “coordinates” in the search space determined by problem variables. As the particle “moves” in the search space, it is also characterised by the current “velocity” v. Every particle is capable of memorising its best position (p), which is where it achieved the best value of the optimisation criterion during the search, as well as the global best position achieved by all particles in the swarm (g). During the search, in the kth iteration, the velocity and the position of each particle are calculated as [50]
(1)$$v^{\left(k+1\right)}=w\cdot v^{\left(k\right)}+cp\cdot rp^{\left(k\right)}\cdot \left(p^{\left(k\right)}-y^{\left(k\right)}\right)+cg\cdot rg^{\left(k\right)}\cdot \left(g^{\left(k\right)}-y^{\left(k\right)}\right),\;y^{\left(k+1\right)}=y^{\left(k\right)}+v^{\left(k+1\right)},$$
where cp and cg are parameters that determine how the personal (local) and social (global) experiences of individuals impact the velocity direction and intensity. The inertia factor w prevents the particles from dispersing in the search space and keeps them together in the swarm, while rp and rg are randomly chosen, mutually independent numbers, uniformly distributed in the range $\left[0,\;1\right]$, which provides the stochastics of the search.

Equation (1) is a common form of the PSO algorithm. The terms “position” and “velocity” must not be taken literally: they are introduced only due to the analogy to biological individuals. The velocity actually represents the correction of variables’ values in each iteration.

Due to the popularity of PSO, many researchers have recently presented different modifications of the original algorithm. They are based on different proposals and suggestions for alterations of parameters cp, cg and w. One of these modifications is called the generalised PSO algorithm (GPSO) [51], which recognises the swarm as a second-order mechanical system, defined as
(2)$$y^{\left(k+1\right)}-2\zeta \rho y^{\left(k\right)}+\rho^{2}y^{\left(k-1\right)}=\left(1-2\zeta \rho +\rho^{2}\right)\left(c\cdot p^{\left(k\right)}+\left(1-c\right)\cdot g^{\left(k\right)}\right),$$
which represents a different form of Equation (1), as is common in control theory [52]. The genuine PSO parameters are replaced by parameters of system dynamics (cp, cg, w → ρ, ζ, c). This form of the algorithm enables more direct control of the dynamics of particle movement in the search space, and it is explained in more detail in [51,53]. This particular modification is used in this paper. It is, however, more suitable to use the iterative form
(3)$$y^{\left(k+1\right)}=\left(1-2\zeta \rho +\rho^{2}\right)\left(c\cdot p^{\left(k\right)}+\left(1-c\right)\cdot g^{\left(k\right)}\right)+2\zeta \rho y^{\left(k\right)}-\rho^{2}y^{\left(k-1\right)},$$
to update the particle positions during the optimisation process.

One can notice that both GA and GPSO are unconstrained optimisation techniques. In order to solve problems with constraints, these algorithms must be adapted to take constraints into account. In this paper, this step is carried out using the penalty function method, which is described in more detail in the subsequent section.

### 2.2. Generalised Robust Estimation of Deformation from Observation Differences

The deformation analysis procedure in the GREDOD method takes place in two phases and can be schematically presented as follows:$$\begin{array}{cc} \underbrace{\Delta l,P_{\Delta }\overset{\mathrm{Robust}\;M-\mathrm{estimation}}{\to }\hat{d},Q_{\hat{d}}} & \underbrace{\hat{d},Q_{\hat{d}}\overset{{\mathrm{Fisher}}^{\prime }s\;\mathrm{test}}{\to }\;\left\{\begin{array}{c} \;T_{i}<F\\ \mathrm{or}\\ \;T_{i}\geq F \end{array}\right.}\\ \mathrm{First}\;\mathrm{phase} & \mathrm{Second}\;\mathrm{phase} \end{array}$$
where $\Delta l=l_{2}-l_{1}$ is the vector of observation differences, $P_{\Delta }={\left(P_{1}^{-1}+P_{2}^{-1}\right)}^{-1}$ is the weight matrix of observation differences, $\hat{d}$ is the estimated displacement vector of network points and $Q_{\hat{d}}$ is a cofactor displacement matrix. The first phase involves the robust estimation of the displacement vector and the corresponding cofactor matrix from the difference in unadjusted observations, while the second phase involves the stability analysis of the network points using Fisher’s test of statistical significance.

The GREDOD method is based on a deformation model in which the vector of the observation differences from two measurement epochs is actually an observation vector, and the displacement vector is a vector of unknown parameters [18,20]:(4)$$v_{\Delta }=Ad-\Delta l,$$
(5)$$\Delta l\;\sim \;N\left(Ad,C_{\Delta }\right),$$
where $v_{\Delta }=v_{2}-v_{1}$ is the vector of the residuals of differences in observations, $\Delta l=l_{2}-l_{1}$ is the vector of observation differences, $A=A_{1}=A_{2}$ is the design matrix, $d=x_{2}-x_{1}$ is the displacement vector, $C_{\Delta }=\sigma_{0}^{2}P_{\Delta }^{-1}$ is the covariance matrix of observation differences and $\sigma_{0}^{2}$ is the *a priori* variance factor.

It is obvious that the displacement vector d in non-free geodetic networks can be determined in a trivial way by using the least-squares method, because the datum defect problem does not exist. On the other hand, in free geodetic networks, the datum defect problem exists, so the functional model (4) has an infinite number of solutions. In order to determine the unique solution of this system in the deformation model, defined by expressions (4) and (5), the following objective functions are integrated:(6)$$v_{\Delta }^{T}P_{\Delta }v_{\Delta }=min,$$
(7)$$\rho \left(d\right)=min,$$
where $\rho \left(\cdot \right)$ is any objective function from the robust M estimation class. The stated objective functions are derived from the corresponding stochastic models [19]. The deformation model defined by expressions (4) and (5) and objective functions (6) and (7) form the optimisation problem of the GREDOD method.

Based on the optimisation condition (6) and the functional model (4), a system of normal equations is obtained:(8)$$N_{\Delta }d-n_{\Delta }=0,$$
where $N_{\Delta }=A^{T}P_{\Delta }A$ and $n_{\Delta }=A^{T}P_{\Delta }\Delta l$ [18,19]. This system of normal equations represents a set of constraints for the displacement vector d. Therefore, when solving the optimisation problem of the GREDOD method, it is necessary to determine the minimum of objective function (7), taking into account the constraints from expression (8). This optimisation problem can be solved by using the Lagrangian multiplier method, from which the following equation system is obtained:(9)$$W\hat{d}-N_{\Delta }M^{-}n_{\Delta }=0,$$
where $W=diag\left(\ldots ,\;w_{i},\ldots \right)$ is a diagonal weight matrix of the displacement vector, $M=N_{\Delta }W^{-1}N_{\Delta }$, and $M^{-}$ is generalised inversion such that $MM^{-}M=M$ holds [19]. Since the datum defect problem is present in free geodetic networks, the design matrix A is of incomplete rank. This is why matrix M is singular, and there is no ordinary inverse of $M^{-1}$.

The generalised inversion of matrix M is determined by the bordered matrix method. Matrix M is bordered by matrix B, after which the generalised inversion is determined [54]:(10)$$G=\left[\begin{array}{cc} M & B\\ B^{T} & 0 \end{array}\right]\;\to \;G^{-}=\left[\begin{array}{cc} Q^{-1}MQ^{-1} & Q^{-1}B\\ B^{T}Q^{-1} & 0 \end{array}\right],$$
where $Q=M+BB^{T}$. It is important to note that the columns of matrix B are independent of the rows of the design matrix A, which results in the independence of the rows of matrix M from the columns of matrix B. Based on expressions (9) and (10), the following expression can be written to estimate the displacement vector:(11)$$\hat{d}=W^{-1}N_{\Delta }M^{-}A^{T}P_{\Delta }\Delta l,$$
where $M^{-}={\left(N_{\Delta }W^{-1}N_{\Delta }+BB^{T}\right)}^{-1}N_{\Delta }W^{-1}N_{\Delta }{\left(N_{\Delta }W^{-1}N_{\Delta }+BB^{T}\right)}^{-1}$. Since the geodetic network datum is defined by network PRPs, the elements of the weight matrix W that refer to the object points must have very small values close to zero. Accordingly, the weight matrix W has the following shape: (12)$$W=diag\left(\ldots ,\;w_{PRP,\;i},\ldots \;\ldots ,\;w_{O,\;i},\ldots \right),$$
where $w_{PRP,i}=w\left({\hat{d}}_{PRP,i}\right)$ is any weight function from the robust M estimation class, and $w_{O,\;i}$ is a small constant value close to zero [19]. Since the displacements in $w\left({\hat{d}}_{PRP,i}\right)$ are unknown, equation system (11) cannot be solved directly. However, the solution can be reached by applying optimisation procedures, such as the IRLS method, the Newton–Raphson method, GA, the GPSO algorithm and similar approaches.

In the case of applying the IRLS method, the numerical solution is formulated as follows:
(13)$${\left.\begin{array}{c} {\hat{d}}^{\left(k\right)}=R^{\left(k\right)}\Delta l\\ Q_{\hat{d}}^{\left(k\right)}=R^{\left(k\right)}{\left(P_{\Delta }\right)}^{-1}{\left(R^{\left(k\right)}\right)}^{T}\\ W^{\left(k+1\right)}=diag\left(\ldots ,\;w_{PRP,i}^{\left(k+1\right)},\ldots \;\ldots ,\;w_{O,\;i},\ldots \right) \end{array}\right\}}_{k=1,\;2,\ldots }$$
where $R^{\left(k\right)}={\left(W^{\left(k\right)}\right)}^{-1}N_{\Delta }{\left(N_{\Delta }{\left(W^{\left(k\right)}\right)}^{-1}N_{\Delta }+BB^{T}\right)}^{-1}N_{\Delta }{\left(W^{\left(k\right)}\right)}^{-1}N_{\Delta }{\left(N_{\Delta }{\left(W^{\left(k\right)}\right)}^{-1}N_{\Delta }+BB^{T}\right)}^{-1}A^{T}P_{\Delta }$, and k is the iteration number. In the first iterative step (k=1), the weight matrix is the identity matrix W=I. In the following iterations, the weights of PRPs $w_{PRP,i}^{\left(k+1\right)}$ are formed by the corresponding weight function, while for the weights of the object points $w_{O,\;i}$, small constant values close to zero are adopted. The iterative procedure (13) is repeated cyclically until the differences between successively transformed displacement vectors $\left|{\hat{d}}^{\left(k+1\right)}-{\hat{d}}^{\left(k\right)}\right|$ become less than the adopted tolerance value γ. Since in this paper, the L1 norm of the displacement vector is used as objective function (7),
(14)$$\rho \left(d\right)=∥d_{1}∥=\sum \left|d_{i}\right|=min,$$
the weight function is of the following shape:(15)wPRP,ik+1=1d^ik+c,
where c is a small constant value that prevents the occurrence of zero in the denominator.

In order to apply the GA and GPSO algorithms to solve the optimisation problem of the GREDOD method—more precisely, equation system (11)—it is necessary to define variables, the feasible search region and the objective function. It is evident that the weights of network PRPs, i.e., network datum points, are variables. In this sense, individual y can be defined as a weight vector of the network PRPs [22]:(16)$$y=\left[w_{PRP,1}\;,\;w_{PRP,2\;},\;\ldots w_{PRP,n}\right].$$

For each individual y in the population, based on its weight values $w_{PRP,i}$, a corresponding weight matrix W can be formed using expression (12). After that, the estimated displacement vector $\hat{d}$ according to expression (11) and the value of objective function (14), which actually represents the individual’s quality, are determined.

It should be borne in mind that very large weight values of PRPs $w_{PRP,i}$ can cause numerical instability when determining the estimated displacement vector $\hat{d}$. In addition, the weight values $w_{PRP,i}$ must be strictly greater than zero ($w_{PRP,i}>0$). In this context, the following constraint is defined:(17)$$w_{min}\leq w_{PRP,i}\leq \;w_{max},$$
where $w_{min}=w_{O,i}$ and $w_{max}=1/c$. This restriction defines the feasible search region. If applying the IRLS method, the weights $w_{PRP,i}$ are determined according to expression (15), so the constraint defined in (17) is always satisfied. On the other hand, the GA and GPSO algorithms do not use expression (15) to determine the weights $w_{PRP,i}$ but start from a series of randomly selected potential solutions and iteratively improve them during the search process. Therefore, the solution obtained by applying these algorithms may be optimal in terms of the value of objective function (14) but unachievable from the physical constraint point of view (17).

For this reason, these algorithms must be modified in some way to be able to solve the constrained optimisation problem, i.e., to allow the search of only those solutions that meet the defined constraint (17). Accordingly, objective function (14) is modified by the penalty function method as follows:(18)$$\rho \left(d\right)=\sum \left|d_{i}\right|+\overset{n}{\underset{i=1}{\sum }}\beta \cdot g_{i}=min,$$
where
$$g_{i}=\left\{\begin{array}{c} \left|w_{PRP,i}-w_{max}\right|,\;w_{PRP,i}>w_{max}\\ 0,\;w_{min}\leq w_{PRP,i}\leq \;w_{max}\\ \left|w_{min}-w_{PRP,i}\right|,\;w_{PRP,i}<w_{min} \end{array}\right.,$$
is the penalty function, and β is the weight coefficient, usually set as a large numerical value to successfully suppress infeasible solutions [55,56]. The penalty functions $g_{i}$ reduce the quality of solutions that exceed the constraint (17), taking into account the distance of the obtained solution from the feasible search region. Accordingly, it is evident that the final solution, i.e., the optimal solution, will be within the defined limits.

The initial step when using the GA is to create the initial population, i.e., in this case, to randomly initialise a starting set of values for weights $w_{PRP,i}$. The initial population is then subjected to an iterative process of transformation, defined by genetic operators, selection, crossover and mutation, in order to improve the quality of generations of individuals in terms of the objective function (fitness), as explained in Section 2.1. The iterative transformation is conducted until one of the stopping criteria (total iteration number, tolerance, accuracy or calculation time) is fulfilled. At the end of the process, the individual providing the best value of objective function (18) is adopted as the optimal solution. Using its weight values $w_{PRP,i}$, a transformation matrix R is formed, which enables the determination of the estimated displacement vector $\hat{d}$ and the corresponding cofactor matrix $Q_{\hat{d}}$. 

The GPSO algorithm also starts from a set of randomly selected individuals (particles) that constitute the initial population (swarm). According to the concept of the GPSO algorithm, the weight values $w_{PRP,i}$ determine the position of each individual in the search space. In order to find the optimal weight values $w_{PRP,i}$, i.e., the optimal datum solution of the displacement vector, the individuals are iteratively “repositioned” in the search space using the approach explained in the previous section. After meeting the adopted stopping criterion, the individual with the best value of the objective function at the population level represents the optimal solution. Finally, based on the weight values $w_{PRP,i}$ of this individual, the estimated displacement vector $\hat{d}$ and the cofactor displacement matrix $Q_{\hat{d}}$ are determined.

In order to make a clear distinction between the estimated displacement vector of individual network points that are the result of measurement errors and those that are the result of actual displacement, it is necessary to analyse the stability of network points using Fisher’s test of statistical significance. In order to apply the Fisher test, the following hypotheses are established:(19)$$H_{0}:E\left({\hat{d}}_{i}\right)=0\;\mathrm{versus}\;H_{a}:E\left({\hat{d}}_{i}\right)\neq 0,$$
where ${\hat{d}}_{i}$ is the estimated displacement vector of the ith point. The procedure for testing the stability of network points is performed as follows:(20)$$T_{i}=\frac{{\hat{d}}_{i}^{T}Q_{{\hat{d}}_{i}}^{-1}{\hat{d}}_{i}}{h_{i}{\hat{\sigma }}_{0}^{2}}\;\sim \;F_{1-\alpha_{0},h_{i},f}$$
where $Q_{{\hat{d}}_{i}}$ is the cofactor displacement matrix of the ith point, ${\hat{\sigma }}_{0}^{2}=\left({\hat{v}}_{\Delta }^{T}P_{\Delta }{\hat{v}}_{\Delta }\right)/f$ is the *a posteriori* variance factor, $h_{i}=rank\left(Q_{{\hat{d}}_{i}}\right)$, $\alpha_{0}=1-{\left(1-\alpha \right)}^{1/m}≅\alpha /m$ is the local significance level, f=n−u+de is the number of degrees of freedom, n is the number of observations, u is the number of unknown parameters, de is the defect datum of the geodetic network, α is the global significance level and m is the number of points [4,5,7]. If $T_{i}<F_{1-\alpha_{0},h_{i},f}$, the null hypothesis $H_{0}$ is not rejected; i.e., the point is declared to be stable. If $T_{i}\geq F_{1-\alpha_{0},h_{i},f}$, the null hypothesis $H_{0}$ is rejected; i.e., the point is declared to be unstable.

It should be noted that the Bonferroni equation $\alpha_{0}=1-{\left(1-\alpha \right)}^{1/m}≅\alpha /m$, which was used in the test (20), neglects the correlation between test statistics, and therefore, the user has no real control of the type I error. Consequently, some authors [21,57,58] have proposed numerically estimating the critical values (quantile values) of the statistical test using the Monte Carlo method. However, the authors consider the Bonferroni method to be effective enough to provide satisfactory results in this research.

### 2.3. Study Area: Šelevrenac Dam

The experimental research in this paper was conducted on an example of a three-dimensional geodetic network for displacement and deformation monitoring of an embankment dam, shown in Figure 1, located on Lake Šelevrenac near the municipality of Inđija in the Republic of Serbia. The length and width of the dam crown are 260.5 m and 6 m, respectively, and the construction height is 13.9 m. The spillway of the dam, located on its left side, is made of reinforced concrete and has a rectangular shape.

The Šelevrenac dam belongs to the group of large dams according to the criteria of the International Commission on Large Dams, because the volume of the lake is higher than one million cubic meters. In geographic coordinates, the dam is located at 45.069937 degrees north latitude and 19.997567 degrees east longitude.

The geodetic network of the Šelevrenac dam consists of 7 PRPs (101–107) and 34 object points (1–38), as shown in Figure 2. In order to obtain quality information about the displacements and deformations of this dam, the observations in this network are realised in time intervals of six months. Each epoch of measurement consists of n=555 observations (185 horizontal directions, 185 zenith angles and 185 slope distances). Observations of horizontal directions, slope distances and zenith angles are performed with standard deviations $\sigma_{\alpha }=1\prime \prime$, $\sigma_{d}=1\;\mathrm{mm}+1.5\;\mathrm{ppm}$ and $\sigma_{Z}=1\prime \prime$, respectively. The number of unknown parameters is u=137 (123 unknown coordinates and 14 unknown orientations), and the defect datum of the geodetic network is de=4. The number of degrees of freedom (f=n−u+de) and the mean redundancy are 422 and 0.76, respectively.

## 3. Results and Discussion

The primary goal of the experimental research presented in this paper was to analyse the efficiency of the proposed modification of the GREDOD method. This modification is based on the application of the GA and GPSO algorithms in the robust estimation of the displacement vector. The deformation analysis procedure is considered efficient if all displaced points of the geodetic network are identified as unstable, and all undisplaced points are identified as stable [6,20]. It is generally known that the efficiency analysis of deformation analysis methods cannot be based on only one set of real observations consisting of two or more measurement epochs in a geodetic network, because in that case, the efficiency analysis refers to only one model of the geodetic network, one set of random measurement errors and one of a multitude of possible displacement and deformation scenarios. In addition, the fact that, in this case, it is not known which network points are really displaced clearly shows that the efficiency of the deformation analysis methods cannot be analysed. Accordingly, their efficiency was analysed by using Monte Carlo simulations and applying MSR. Here, efficiency conclusions were drawn based on a large number of simulated observation sets representing different displacement and deformation scenarios in the geodetic networks that are the subject of analysis. The procedures for generating simulated observation sets and efficiency analysis of deformation analysis methods are discussed in detail in the literature [6,11,13,20,22]. 

The efficiency analysis of the proposed modification of the GREDOD method was performed on a test sample generated for the needs of experimental research presented in [22], which consists of 120,000 simulated observation sets in the geodetic network for displacement and deformation monitoring of the Šelevrenac embankment dam. This test sample includes the following three variants of object points displacement:
Variant 1—one randomly selected object point is displaced ($n_{O}=1$);Variant 2—two randomly selected object points are displaced ($n_{O}=2$);Variant 3—three randomly selected object points are displaced ($n_{O}=3$).

Each of the defined variants of object points displacement includes eight different cases of network PRPs displacement:
All PRPs are undisplaced ($n_{PRP}=0$);One randomly selected PRP is displaced ($n_{PRP}=1$);Two randomly selected PRPs are displaced ($n_{PRP}=2$);Three randomly selected PRPs are displaced ($n_{PRP}=3$);Four randomly selected PRPs are displaced ($n_{PRP}=4$);Five randomly selected PRPs are displaced ($n_{PRP}=5$);Six randomly selected PRPs are displaced ($n_{PRP}=6$);All PRPs are displaced ($n_{PRP}=7$).

It is important to note that the magnitudes of the displacement vectors of PRPs and object points ($s_{PRP,i}$ and $s_{O,i}$) take values from the interval $\left[r_{i},2r_{i}\right]$, where $r_{i}$ is the radius of the displacement sphere whose volume is equal to the volume of the corresponding displacement ellipsoid [22]. For each of the eight previously mentioned cases of PRPs displacement, 5000 simulated observation sets were generated within all three variants of object points displacement. It should also be noted that the observations were simulated together with random measurement errors that follow a normal distribution with a mean value of zero and the standard deviations for horizontal directions, slope distances and zenith angles listed in Section 2.3.

The test sample was extended with one characteristic displacement scenario where all network points (PRPs and object points) are undisplaced $(n_{D}=0$) in order to examine the false-positive rate (FPR) of the proposed GREDOD modification, as reported in [59], where a similar procedure was conducted for outlier analysis. FPR is defined as the number of false-positive rates divided by the total number of experiments. In this scenario, 5000 simulated observation sets were generated, as explained in [22]. These observations were simulated together with random measurement errors that follow a normal distribution, as explained in the previous paragraph. All simulations were conducted using the Monte Carlo method implemented in the Matlab software package. This test sample is available in [60].

The deformation analysis was performed on each set of simulated observations using the GREDOD method, whereby the GA and GPSO algorithms were applied in addition to the IRLS method in the process of robust estimation of the displacement vector. The values of the IRLS method parameters were adopted based on empirical analysis of the optimisation process. Values of 0.01 mm and 0.001 mm were adopted for the constant c and tolerance γ, while a value of 0.0001 was adopted for the weights of object points $w_{O,\;i}$. 

Based on the adopted values of the IRLS method parameters and expression (17), the constraint $0.0001\leq w_{PRP,i}\leq \;100$ was defined, specifying the feasible search region. This constraint was integrated into objective function (18) by the penalty function method, where the value $10^{6}$ was adopted for the weight coefficient of the penalty β. 

Schemes for setting the parameters of the GA and GPSO algorithms were adopted in accordance with the recommendations in the literature [48,51]. The selection of individuals was performed using stochastic uniform selection with linear ranking, a crossover of individuals using a uniform crossover scheme and gene mutation by simple random change using a normal distribution. It is important to note that the change in generations was carried out by applying an elitist strategy, where 5% of the best individuals from the population are directly transferred to the next generation. In the GPSO algorithm, the parameters ρ and c decrease linearly within the ranges $\left[0.95,\;0.6\right]$ and $\left[0.8,\;0.2\right]$, respectively, during the optimisation process, while the parameter ζ takes values from the range $\left[-0.9,\;0.2\right]$ using a uniform distribution. 

When the population (swarm) size and the stopping criterion are considered, it is generally very difficult to provide a strong recommendation in advance on how to set their values. This characteristic problem is usually solved by applying the trial-and-error method. In this study, for the size of the population (swarm), i.e., the number of individuals (particles), the value 400 was adopted. The stopping criterion is defined by the maximum number of generations (iterations) and tolerance. The values 100 and $10^{-6}$ were adopted for these parameters, respectively. By performing an empirical analysis of the optimisation process convergence, which was conducted on several sets of simulated observations that reflect different displacement and deformation scenarios, it was confirmed that these parameter values provide satisfactory results.

This specifies all parameter values necessary for the application of the IRLS method, GA and GPSO algorithm in the robust estimation of the displacement vector in the GREDOD method. The next phase in the deformation analysis procedure is to examine the stability of the network points using Fisher’s test of statistical significance. For the global significance level α, the value 0.05 was adopted in this test, so the local significance level $\alpha_{0}$ was 0.00125. 

The MSRs of the GREDOD method were independently calculated for each of the eight analysed cases of PRPs displacement in all three variants of object points displacement. The FPRs of the GREDOD and IWST methods were calculated for the scenario where all points (PRPs and object points) are undisplaced. It is important to note that in this paper, the deformation analysis procedure is considered successful if all displaced object points are identified as unstable, and all undisplaced object points are identified as stable. 

The obtained results are directly comparable to the results presented in [22], where the efficiency of the IWST method when applying the IRLS method, GA and GPSO algorithm in the robust estimation of the displacement vector was analysed on the same test sample. Accordingly, in the following, among other things, a comparative analysis of the efficiency of the IWST and GREDOD methods is presented.

The MSRs of the GREDOD method related to the first variant of object point displacement are presented in the form of diagrams on the left-hand side of Figure 3. It is evident that in the case of the IRLS method, the efficiency of the GREDOD method decreases significantly with the increasing number of displaced PRPs $n_{PRP}$, while in the case of applying the GA and GPSO algorithms, the efficiency does not change significantly. In order to explain these conclusions in more detail, two characteristic cases of point displacement in a potential reference network are analysed below. If the first case of displacement is observed ($n_{PRP}=0$), it can be seen that applying all three optimisation procedures (IRLS, GA and GPSO) results in very similar values of MSRs. In addition, the case in which all PRPs are displaced ($n_{PRP}=7$) is also considered. The efficiency of the GREDOD method is 57.82% and 57.58% higher when applying the GA and GPSO algorithms, respectively, than in the case of the IRLS method. The diagram on the right-hand side of Figure 3 presents MSRs obtained using the IWST method in [22], presented for reference and comparison purposes. It can be observed that the efficiency of the IWST and GREDOD methods is very similar in all eight cases of PRPs displacement. Differences in efficiency between the IWST and GREDOD methods in the cases of the IRLS method, GA and GPSO algorithms range from 0.02% to 0.60%, from 0.04% to 0.50% and from 0.02% to 0.80%, respectively.

Figure 4 presents the MSRs of the GREDOD method and refers to the second variant of object points displacement. It is obvious that when applying the GA and GPSO algorithms, the GREDOD method efficiency does not change significantly with an increasing number of displaced PRPs $n_{PRP}$, while in the case of the IRLS method, its efficiency decreases significantly. These conclusions are fully consistent with the conclusions derived from the analysis of the results obtained for the first variant ($n_{O}=1$). However, it should be noted that for this variant, the GREDOD method efficiency is about 6% lower on average compared to the previous variant. This diagram (Figure 4, right) also shows the MSRs of the IWST method obtained in [22]. It is evident that the efficiency of the IWST and GREDOD methods is very similar in all eight cases of point displacement in a potential reference network.

The MSRs of the GREDOD and IWST methods related to the third variant, where three randomly selected object points are always displaced, are shown in the diagram in Figure 5. All previously drawn conclusions related to the GREDOD method efficiency behaviour when applying the IRLS method, GA and GPSO algorithm are also valid for this variant of object points displacement. It is important to note that the GREDOD method efficiency is about 11% lower on average compared to the first variant. In addition, it is seen that for this variant of object points displacement, the efficiency of the IWST and GREDOD methods is very close in all eight cases of PRPs displacement. The MSRs of the IWST method are taken from [22].

Based on the values of the MSRs, the overall efficiency values of the GREDOD method were calculated for all three variants of object points displacement. The overall efficiency is defined as the arithmetic mean value of the MSRs related to individual cases of PRPs displacement. The overall efficiency values of the GREDOD and IWST methods are presented in the form of diagrams in Figure 6, where the overall efficiency values of the IWST method are taken from Reference [22]. It is obvious that the GREDOD method efficiency decreases with an increasing number of displaced object points $n_{O}$ when applying all three optimisation procedures. This problem can be successfully solved using a strategy based on dividing the geodetic network into as many subnetworks as there are object points, where each subnetwork consists of all PRPs and only one object point [8,13]. In addition, it is seen that the GREDOD method efficiency is significantly improved by applying the GA and GPSO algorithms in the robust estimation of the displacement vector. The improvement percentages of the overall efficiency of the GREDOD method range from 18.24% to 18.65% with the genetic algorithm and from 17.76% to 17.97% with the GPSO algorithm. It is important to point out that these values represent relative increases in the GREDOD method efficiency in relation to the efficiency obtained by applying the IRLS method. In addition, it can be seen that the values of the overall efficiency of the IWST and GREDOD methods are very close for all three variants of object points displacement.

The FPRs of the GREDOD and IWST methods related to the displacement scenario where all network points are undisplaced ($n_{D}=0$) are shown in the diagram in Figure 7. It can be observed that the FPRs of the GREDOD and IWST methods are slightly higher with the IRLS method than with the GA and GPSO algorithms. The FPR of the GREDOD method is 0.40% lower for GA and 0.38% lower for the GPSO algorithm compared to the IRLS method application. The FPR of the IWST method is 0.56% lower for GA and 0.68% lower for the GPSO algorithm.

Additional analysis of the obtained results was conducted based on the values of the overall absolute true errors of the estimated displacement vectors, determined by using all three optimisation algorithms (IRLS, GA and GPSO) with the GREDOD method, for each set of simulated observations. The overall absolute true error of the estimated displacement vector is defined as
(21)$$e_{\hat{d}}=\sum \left|d_{i}^{s}-{\hat{d}}_{i}\right|,$$
where $d_{i}^{s}$ and ${\hat{d}}_{i}$ are components of the simulated and estimated displacement vectors, respectively. Figure A1, Figure A2 and Figure A3 (Appendix A) present the empirical distributions of these errors, which refer to the first, second and third variants of object points displacement, respectively. Arithmetic mean values and standard deviations of the mentioned errors were independently calculated for each of the eight cases of PRPs displacement within all three variants of object points displacement.

Figure 8 depicts the arithmetic mean values and standard deviations of the overall absolute true errors of the estimated displacement vectors of the GREDOD method when one randomly selected object point is displaced (variant 1). When the GA and GPSO algorithms are applied, one can note that there is no significant change in the arithmetic mean values or standard deviations of these errors with an increasing number of displaced PRPs, $n_{PRP}$. On the other hand, when the IRLS method is applied, these values evidently increase as $n_{PRP}$ increases. The right-hand side of Figure 8 depicts the same values for the IWST method. These values are taken from [22] in order to make a comparison of these two methods. It is notable that arithmetic mean values and standard deviations of the overall absolute true errors of the estimated displacement vectors are almost identical between the IWST and GREDOD methods.

The arithmetic mean values and standard deviations of the overall absolute true errors of the estimated displacement vectors of the IWST and GREDOD methods related to the second and third variants of the object points displacement are shown in the form of diagrams in Figure 9 and Figure 10 respectively. The arithmetic mean values and standard deviations of these errors of the IWST method are taken from [22] for the purpose of comparative analysis of the results obtained using the IWST and GREDOD methods. Since the arithmetic mean values and standard deviations of these errors are very similar for all three variants of object points displacement, all previously drawn conclusions regarding the behaviour of these errors in the case of the IRLS method, GA and GPSO algorithm are valid for these two variants of object points displacement. In addition, it is seen that the behaviour of the overall true errors of the estimated displacement vectors is consistent with the efficiency behaviour of the IWST and GREDOD methods.

In addition to the above-presented results, the computational effort needed to apply the described methods must also be considered. The calculation time for both the GA and GPSO algorithms is longer compared to the IRLS method. The calculation time for the GA and GPSO algorithms is directly related to parameter values such as population size and stopping criteria. Comparing the GA and GPSO algorithms, it is notable that the GPSO algorithm works faster, because the calculations are simpler.

A general conclusion regarding computational effort is that GA is slightly more efficient than the GPSO algorithm, but it demands more calculation effort. Therefore, it is more suitable to use GA when time is not crucial. For some real-time applications, the GPSO algorithm is generally a much better choice.

## 4. Conclusions

This paper presents an original modification of the GREDOD method based on the application of two evolutionary optimisation algorithms, the GA and GPSO algorithms, in the robust estimation process of the displacement vector. In this procedure, the optimal weight values of the PRPs are determined. This is achieved by solving the optimisation problem formed by the deformation model defined by expressions (4) and (5) and objective functions (6) and (7). Since the weight values illustrate the contribution of each PRP to the datum definition of the displacement vector, the optimisation problem of the GREDOD method can be treated as a problem of determining the optimal datum of the displacement vector. For the purpose of solving this optimisation problem, the IRLS method is traditionally applied. This method is based on the iterative improvement of one initial solution obtained by the least-squares method. The main disadvantage of this method is that it is not possible to determine a global optimal solution to the optimisation problem of the GREDOD method if displaced points appear in the set of PRPs. Accordingly, the application of the GA and GPSO algorithms in the process of robust estimation of the displacement vector is proposed. Unlike the IRLS method, the mentioned algorithms start from a series of randomly selected potential solutions (individuals or particles), which distribute the initial population (swarm) throughout the whole search space, and transform them in a controlled random manner in order to find a (near) global optimal solution for the optimisation problem. Specifically, the use of randomness allows these algorithms to “jump out” of the local optimum in order to find the global optimum. In order to apply these algorithms, the individual (particle) is defined as the weight vector of the PRPs (16), and the feasible search region is specified by (17), i.e., a constraint on the weight of PRPs, which is integrated into objective function (18) by the penalty function method. Despite being very simple, this form of penalty function has proven to be effective enough to provide quite satisfactory results in this research.

In the experimental research, a very thorough and exhaustive efficiency analysis of the application of the IRLS method, GA and GPSO algorithm in the robust estimation process of the displacement vector in the GREDOD method was performed. The efficiency of the previously mentioned optimisation procedures was analysed by applying MSR to a test sample of 125,000 simulated observation sets in the geodetic network for displacement and deformation monitoring of the embankment of the Šelevrenac dam. A comparative analysis was performed with the results obtained using a modified IWST method in which the optimisation problem is solved using the GA and GPSO algorithms [22]. Based on the analysis of the research results, conclusions were drawn on the behaviour of the efficiency, FPRs and overall absolute true errors of the estimated displacement vector by applying the GREDOD method in the case of the IRLS method, GA and GPSO algorithm. In addition, the main advantages of evolutionary procedures compared to the classical IRLS method were highlighted. A global conclusion emerges from this analysis. The GREDOD method efficiency was significantly improved by applying the GA and GPSO algorithms in the process of robust estimation of the displacement vector, i.e., in the process of determining the optimal datum of the displacement vector. In this context, it can be concluded that by applying these algorithms, the error of determining the optimal datum of the displacement vector is significantly reduced, which increases the degree of robustness of the GREDOD method to the existence of displaced PRPs. Based on these facts, it can be concluded that the reliability of the deformation analysis results is also improved, which is very valuable for practical applications in geodetic monitoring of engineering facilities and parts of the Earth’s crust surface.

There are several possibilities for future research. In particular, evolutionary optimisation algorithms, such as GA, GPSO or other similar techniques, can be applied to solve optimisation problems of the RDA methods based on M_split_ estimation. In addition, the possibility of additional improvement of the efficiency of evolutionary algorithms in solving the optimisation problem of the RDA methods, such as IWST and GREDOD methods, by applying some of the advanced forms of penalty functions can be considered. Furthermore, the efficiency analysis should include some different models of geodetic networks for monitoring the deformation and displacement of engineering facilities and the Earth’s crust surface.

## Figures and Tables

**Figure 1 sensors-22-00159-f001:**
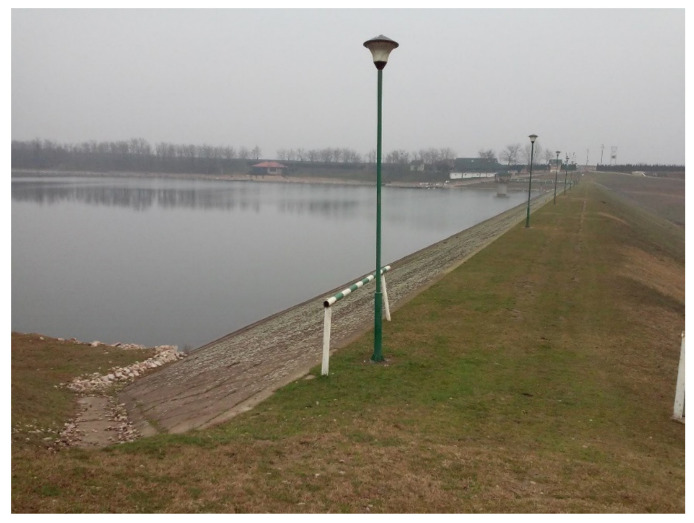
Šelevrenac embankment dam.

**Figure 2 sensors-22-00159-f002:**
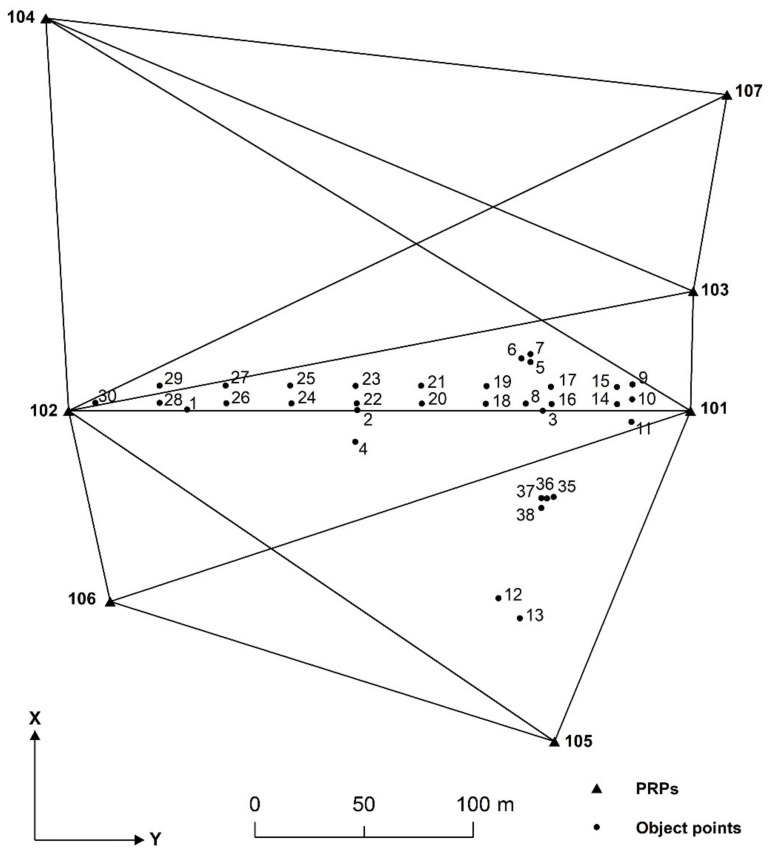
Geodetic network of Šelevrenac dam.

**Figure 3 sensors-22-00159-f003:**
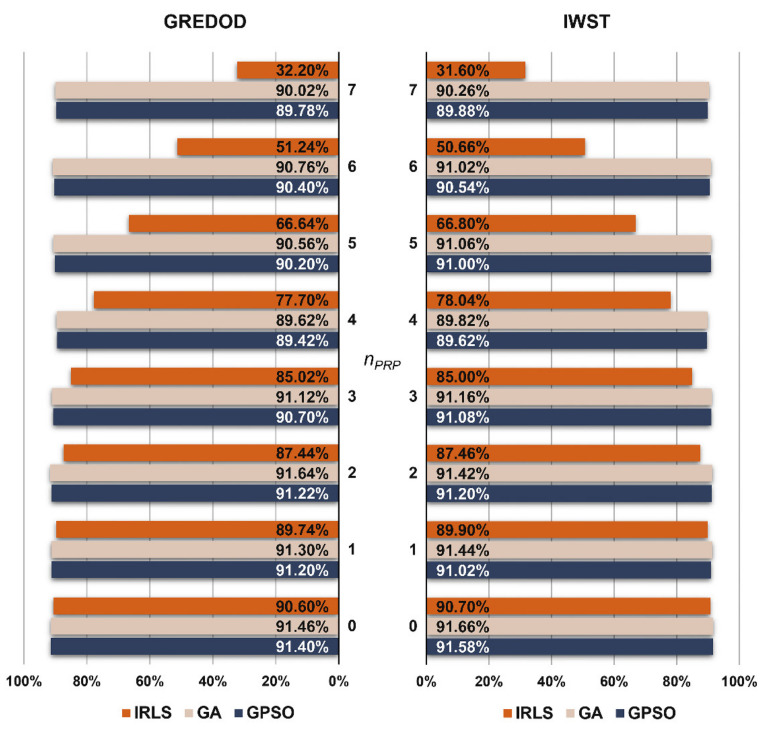
MSRs of GREDOD and IWST methods for variant 1.

**Figure 4 sensors-22-00159-f004:**
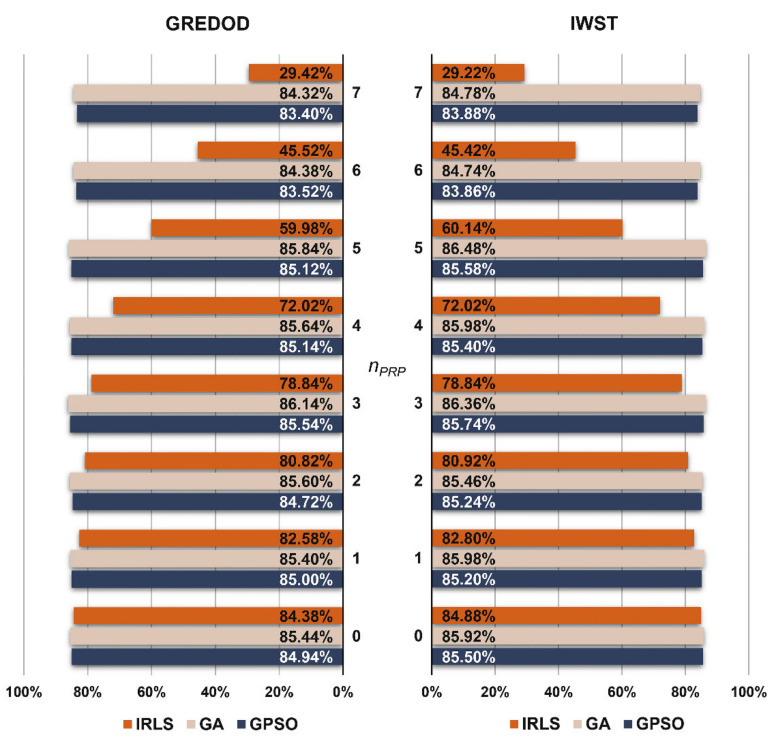
MSRs of GREDOD and IWST methods for variant 2.

**Figure 5 sensors-22-00159-f005:**
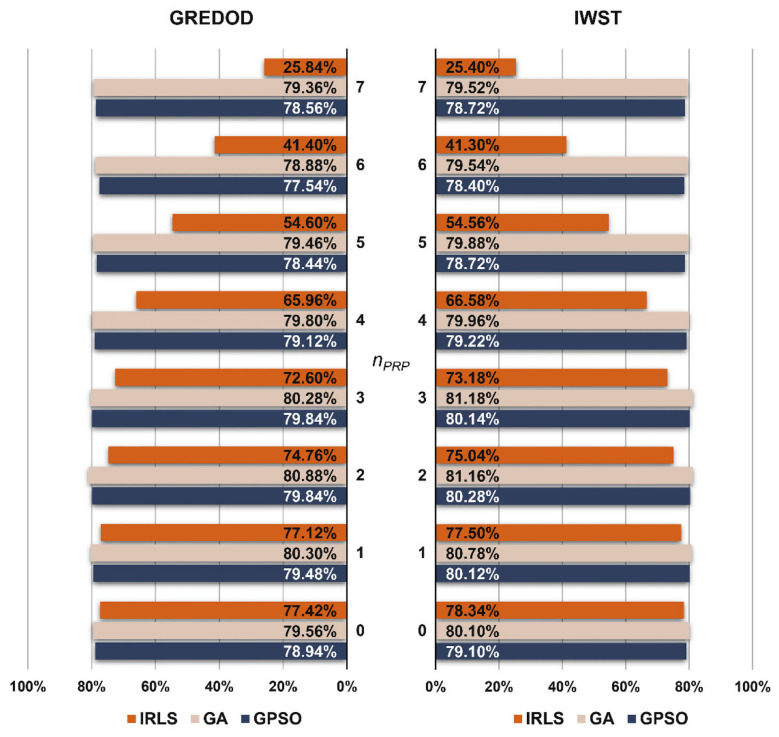
MSRs of GREDOD and IWST methods for variant 3.

**Figure 6 sensors-22-00159-f006:**
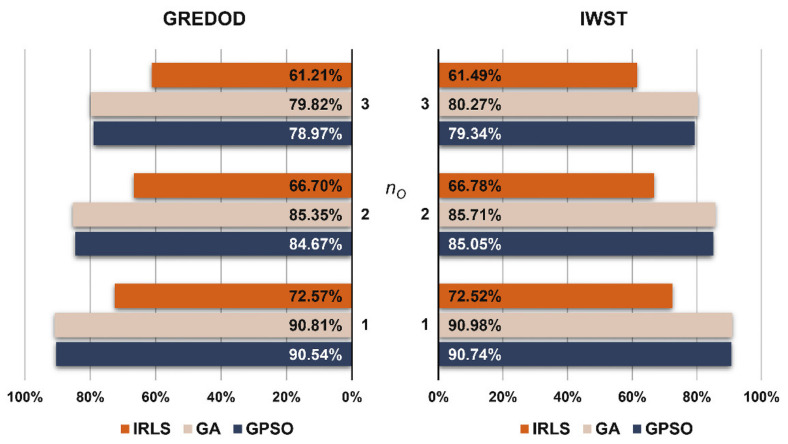
Overall efficiency of GREDOD and IWST methods.

**Figure 7 sensors-22-00159-f007:**
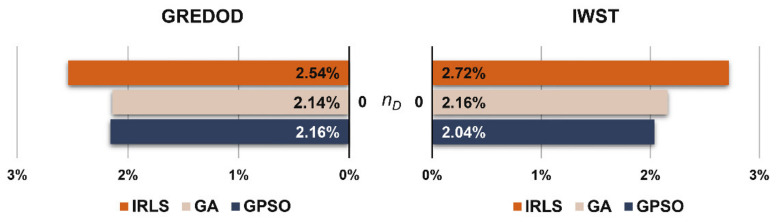
FPRs of GREDOD and IWST methods.

**Figure 8 sensors-22-00159-f008:**
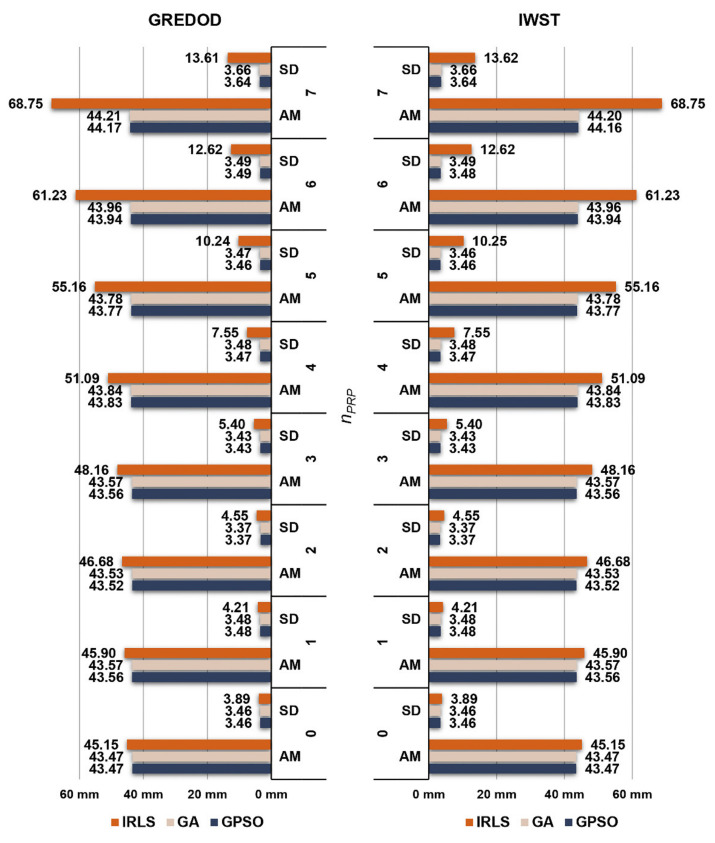
Arithmetic mean (AM) values and standard deviations (SDs) of overall absolute true errors $e_{\hat{d}}$ of the GREDOD and IWST methods for variant 1.

**Figure 9 sensors-22-00159-f009:**
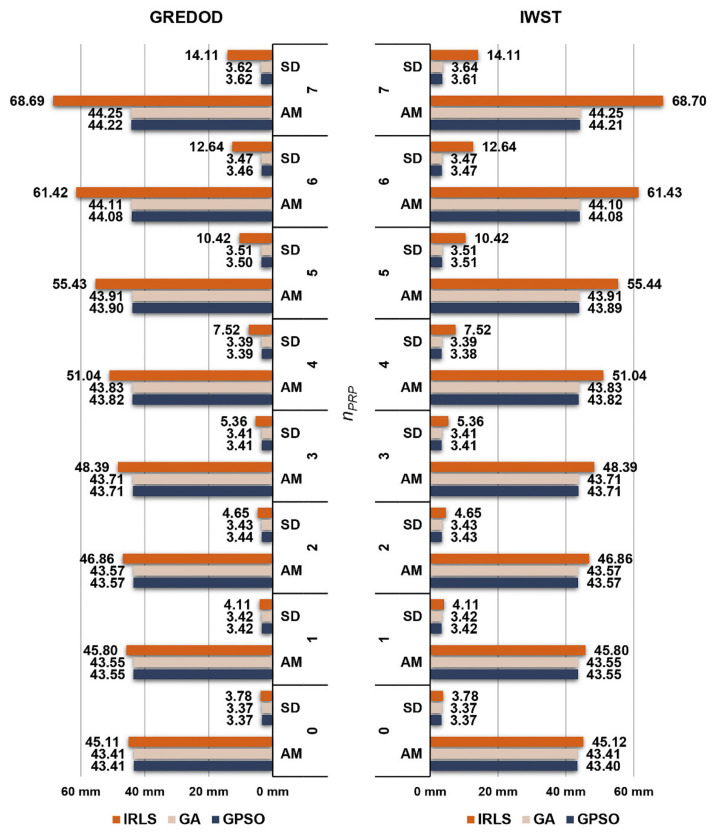
Arithmetic mean (AM) values and standard deviations (SDs) of overall absolute true errors $e_{\hat{d}}$ of the GREDOD and IWST methods for variant 2.

**Figure 10 sensors-22-00159-f010:**
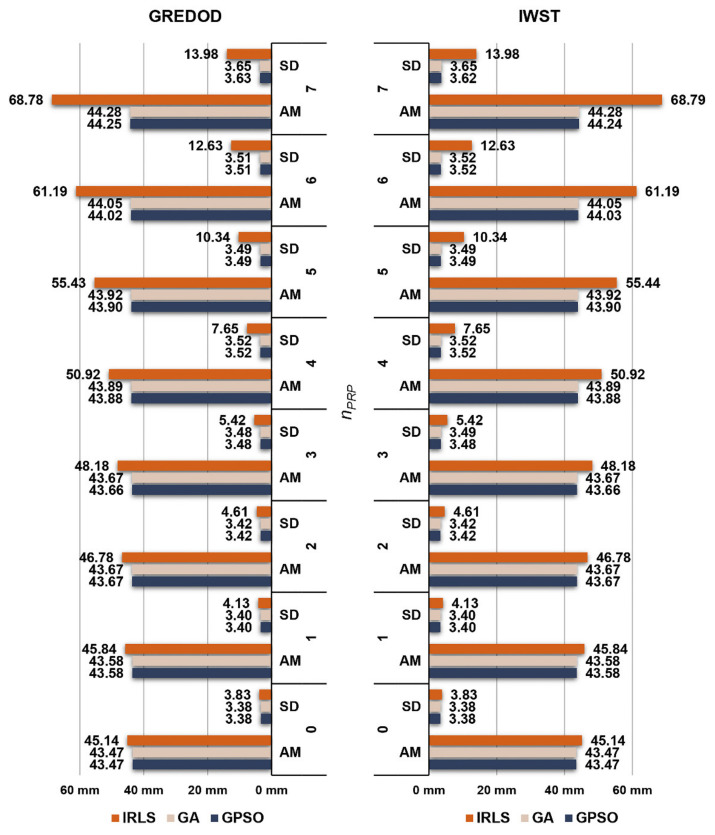
Arithmetic mean (AM) values and standard deviations (SDs) of overall absolute true errors $e_{\hat{d}}$ of the GREDOD and IWST methods for variant 3.

## Data Availability

Test sample and Matlab code for the GPSO algorithm are openly available in Mendeley [60].

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
