# Peer review of "Robust Estimation of Deformation from Observation Differences Using Some Evolutionary Optimisation Algorithms"

_sensors, 2021, doi:10.3390/s22010159_

Round 1

Reviewer 1 Report

Paper: sensors-1423635. Robust estimation of deformation from observation differences using some evolutionary optimisation algorithms.

In this paper, an original modification of generalised robust estimation of  deformation from observation differences (GREDOD) method has been presented, involving the application of two evolutionary optimisation algorithms, genetic algorithm (GA) and generalised particle swarm optimisation (GPSO), in the procedure of robust estimation of the displacement vector.

Comments:

  1. Please include in the abstract the practical usefulness of this study.
  2. Please include in the abstract quantitative information to support the main findings of the study (algorithm used).
  3. Include the main results of this study in the abstract.
  4. Review the keywords used. Some are very extensive.
  5. The paragraphs in the introduction are very long. Please review the wording and style.
  6. In the introduction visualize in a clear and concrete way the objective of this study.
  7. Include in the introduction information about the practical usefulness of this research.
  8. Include references associated with equations 1 and 2.
  9. I suggest including a conventional chapter on materials and methods. That is, where the place of study, the information collection system, and the analysis of the information are visualized. For example, what was the simulation software used in this study?
  10. Include in the chapter on materials and methods a description of the places of study. For example, from Šelevrenac dam.
  11. This paragraph is very long. Review writing and style. L369-408.
  12. Currently the paper is difficult to read due to the existence of very long paragraphs.
  13. Please include a nomenclature section at the end of the paper.
  14. The main limitation of paper is that it is very theoretical. In other words, the study should be based on a couple of application (engineering) case studies. Case studies should be evident throughout the results and discussion chapter. It is very relevant to demonstrate the practical utility of this study.
  15. Please increase the case studies used in this paper.
  16. Finally, a chapter on materials and methods is not obvious. It is important to include it and structure it properly.
  17. Please include statistical tests of group differences to compare the fit of the methods used.
  18. Focus the discussion of results towards practical effects in the places of study.
  19. Today, this paper is more suitable for a journal with a focus on pure mathematics.

Author Response

The authors would like to express their gratitude to the anonymous Reviewer for the insightful comments and remarks. We accept all recommendations and comments of the Reviewer. In the revised manuscript, all significant changes have been marked according to journal’s instructions. Answers and discussions regarding particular issues raised by the Reviewer are given in document Response to Reviewer 1.

Reviewer 2 Report

Dear authors,

My overall impression is very positive. However, I found some of the description of the paper to be too detailed, while the description of some very important points was inadequate or completely missing. I explain my concerns in more detail below.

- An important experiment was missing. Analyze the false positive rates, i.e., apply the methods under conditions where all points (PRPs and object points) are stable. It is very common to only evaluate on detection, but it is important to also check false alarm rates, as can be seen here:

Ivandro Klein, Stefano Sampaio Suraci, Leonardo Castro de Oliveira, Vinicius Francisco Rofatto, Marcelo Tomio Matsuoka & Sergio Baselga (2021) An attempt to analyse Iterative Data Snooping and L1-norm based on Monte Carlo simulation in the context of leveling networks, Survey Review, DOI: 10.1080/00396265.2021.1878338

- page 8, Line 312: You use Bonferroni to control the probability of committing the Type I Error (alfa_0 = alfa/m). However, this approach neglects the correlation between test statistics and therefore the user has no real control of the Type I Error. Although I understand that for a system with high redundancy and low correlation between statistics, Bonferroni is sufficient. Please, it is important to highlight this idea and support it based on the following references:

Rofatto, V.F.; Matsuoka, M.T.; Klein, I.; Roberto Veronez, M.; da Silveira, L.G., Jr. A Monte Carlo-Based Outlier Diagnosis Method for Sensitivity Analysis. Remote Sens. 2020, 12, 860. https://doi.org/10.3390/rs12050860

Lehmann, R. Improved critical values for extreme normalized and studentized residuals in Gauss–Markov models. J Geod 86, 1137–1146 (2012). https://doi.org/10.1007/s00190-012-0569-0

- page 8, Lines 312-313: what is the "n", "u" and "de" ? Is "n" the number of observations from all epochs, "u" the number of parameters (displacements)? I did not understand what "de" means. What does "hi = r(Q_di)" mean? As I understand it would be the local redundancy number? Please explain these expressions better so there is no confusion for the reader.

- page 8, Expression (19): You have applied the one tailed F test. Therefore, the alternative hypothesis should be E(d_i) > 0 instead of E(d_i) ≠ 0.

- page 10, Lines 388-390: "Population (swarm) size and stopping criterion are defined on the basis of empirical analysis of the optimisation process convergence, which is conducted on several sets of simulated observations that reflect different displacement and deformation scenarios." The impression remains that the choice of the size of the swarm population and the stopping criterion depend on prior knowledge of the simulated displacement. Therefore, the way in which the authors conducted the experiments seems to me to be a little biased, since the displacement must be informed in advance so that the GPSO can adjust better. If not, I suggest that authors make these choices clearer to the reader.

- page 9, Line 361-362: What actual values were used to define the magnitude of the displacement?

- It would be important to better detail the geodetic network in the Results and Discussion topic (number of observations, unknown parameters, redundancy, local redundancy number, etc.)

- Why did you use a significance level of 0.05? What would happen to the results if others were used? Please consider the following work to this discussion:

Nowel, K. Specification of deformation congruence models using
combinatorial iterative DIA testing procedure. J Geod 94, 118 (2020).
https://doi.org/10.1007/s00190-020-01446-9

- Success rates for both IWST and GREDOD are roughly equal. It would be important to highlight the computational cost. What would be a lower computational cost? The same question in case of GA versus GPSO.

- It would be important to make a background of the IWST method to make the work easier to read.

- Please provide flowcharts for both the methods and experiments performed.

- Please reduce the size of the paragraphs or break them into others so that the text has a more adequate fluidity.

- Finally, I recommend that authors make the data and scripts available in a publicly accessible repository.

Author Response

The authors would like to express their gratitude to the anonymous Reviewer for the insightful comments and remarks. We accept all recommendations and comments of the Reviewer. In the revised manuscript, all significant changes have been marked according to journal’s instructions. Answers and discussions regarding particular issues raised by the Reviewer are given in document Response to Reviewer 2.

Round 2

Reviewer 1 Report

Comments:

  1. Review the keywords used. Some are very extensive.
  2. Please include an abbreviations section at the end of the paper.

Author Response

The authors would like to express their gratitude to the anonymous Reviewer for the insightful comments and remarks. Answers and discussions regarding particular issues raised by the Reviewer are given in document Response to Reviewer 1.

Reviewer 2 Report

I congratulate the authors for their work. The authors have applied all of my comments. I have no further comments.

Author Response

The Authors would like to express their gratitude to the Reviewer for all his suggestions and comments, which were extremely helpful in the process of paper revision.